# Intraretinal Cysts as a Manifestation of Retinal Angiomatous Proliferation in Optical Coherence Tomography Angiography

**DOI:** 10.3390/medicina58050676

**Published:** 2022-05-19

**Authors:** Jakub J. Kałużny, Przemysław Zabel, Beata Danek, Damian Jaworski, Jarosław Makowski

**Affiliations:** 1Department of Sensory Organ Studies, Collegium Medicum, Nicolaus Copernicus University, 85-067 Bydgoszcz, Poland; jjkaluzny@oftalmika.pl (J.J.K.); przemo.zab@gmail.com (P.Z.); danekbeata@gmail.com (B.D.); 2Oftalmika Eye Hospital, 85-631 Bydgoszcz, Poland; optometrysta_jarek@wp.pl; 3Division of Ophthalmology and Optometry, Department of Ophthalmology, Collegium Medicum, Nicolaus Copernicus University, 85-067 Bydgoszcz, Poland

**Keywords:** intraretinal cysts, retinal angiomatous proliferation, intraretinal neovascularization, chorioretinal anastomosis, SDOCT, OCTA

## Abstract

*Background and Objectives*: Intraretinal cysts are common pathology observed inspectral domain optical coherence tomography (SDOCT) in patients with neovascular form of age-related macular degeneration (AMD). The aim of the study was to determine if the presence of intraretinal cysts is positively correlated with diagnosis of retinal angiomatous proliferation (RAP) in optical coherence tomography angiography (OCTA). *Material and Methods*: A total of 21 eyes with intraretinal cysts in SDOCT exam (Group1) and 21 eyes with subretinal fluid(Group 2) were enrolled into the study. In each eye, the presence of intraretinal neovascularization (IRN) and chorioretinal anastomosis (CRA) was evaluated in OCTA by two experienced graders. *Results*: IRN was observed in 20 eyes (95.2%) from Group 1 and 5 eyes (23.8%) from Group 2. Features of CRA were found in 18 eyes (80.95%) and 16 eyes (76.2%) respectively for Group 1 and 2. Patients with cysts are 50 (95% CI: 5.43–460.52) times more likely to have IRN (*p* < 0.001). *Conclusions*: The presence of intraretinal cysts on SDOCT retinal sections in eyes with neovascular AMD corresponds to the presence of IRN on OCTA examination. The results indicate that the absence of a cyst does not exclude the presence of IRN and CRA which can be identified on OCTA.

## 1. Introduction

Retinal angiomatous proliferation (RAP) also known as type 3 macular neovascularization (MNV3) is one type of exudative age-related macular degeneration (nAMD) [1]. This condition is characterized by the appearance of intraretinal neovascularization (IRN) originating either from deep vascular complexes (DVC) or the choroid [2,3]. MNV3 accounts for 15–20% of all cases of nAMD. Its clinical presentation differs from type 1 macular neovascularization (MNV1) where the pathological vessels are located under the retinal pigment epithelium (RPE) and macular neovascularization type 2 (MNV2) where the neovascular membrane is located under the sensory retina [3,4].

According to the classification proposed by Yannuzzi, stage I of RAP development involves the proliferation of capillaries originating from the DVC resulting in the formation of IRN [1,2,3]. These pathological vessels gradually spread towards the outer layers of the retina. Intraretinal hemorrhages and retinal edema may be seen on fundus examination. In stage II, pathological intraretinal vessels extend deeper into the subretinal space resulting in subretinal neovascularization. In addition, RPE hyperplasia and connections between retinal vessels begin to appear. In stage III of the disease, the pathological vessels under the RPE are observed together with the formation of a chorioretinal anastomosis (CRA) [1].

The diagnosis of MNV3 requires visualization of abnormal vessels within the sensory retina, retinal-retinal anastomoses, or CRA using fluorescein and indocyanine angiography (IA) or optical coherence tomography angiography (OCTA). While, in many cases, the analysis of retinal lesions seen on spectral domain optical coherence tomography (SDOCT) is sufficient to make the initial diagnosis, the final diagnosis should be supported by OCTA. Intraretinal cysts are a typical feature of RAP on SDOCT imaging and are present in nearly 100% of cases [5,6].

Fluid escaping from the pathological IRN vessels can accumulate within the retina, forming cysts during all stages of RAP. These intraretinal cysts can be easily visualized on SDOCT. They appear as oval hyporeflective spaces of varying sizes located within the sensory retina. Therefore, it can be hypothesized that in nAMD patients, the presence of intraretinal cysts on SDOCT is indicative of IRN.

This study aims to determine whether the appearance of intraretinal cysts in patients with nAMD is consistent with the presence of IRN and CRA on OCTA and is sufficient to diagnose MNV3. In addition, this study will evaluate whether the presence of IRN and CRA in the absence of intraretinal fluid is possible.

## 2. Materials and Methods

This retrospective study analyzed the medical records of patients treated for nAMD with anti–vascular endothelial growth factor therapy (anti-VEGF) injections at the Oftalmika Eye Hospital between 2017 and 2020. This study was approved by the Bioethics Committee of Nicolaus Copernicus University Collegium Medicum. Patients over the age of 55 who during the qualifying visit for treatment with anti-VEGF intravitreal injections had a SDOCT revealing the presence of intraretinal cysts and/or subretinal fluid in the macula accompanied by nAMD symptoms in the form of fibrovascular pigment epithelial detachment (fvPED), drusen, or pseudodrusen were included in the analysis. Eyes in which the MNV was caused by a condition other than nAMD and in particular eyes with myopia above −5.0 Diopterswere excluded from the study. Eyes in which the origin of the fluid was not due to nAMD (diabetic retinopathy, central retinal vein, or branchocclusion, central serous chorioretinopathy, macular telangiectasia) were also excluded from the study. Additionally, patients with the presence of epiretinal membranes were also excluded. Subjects with incomplete medical records and opaque optical media were not included in this study.

The qualified eyes were divided into two groups. Group 1 included eyes with the presence of intraretinal cysts on SDOCT regardless of whether subretinal fluid was present. Group 2 consisted of eyes in which the fluid present on SDOCT examination was only under the sensory retina and no retinal cysts were present. Both study groups had examinations performed with SDOCT and OCTA prior to treatment.

Macular structure assessment was conducted by SDOCT (Spectralis OCT, Heidelberg Engineering, Germany) using the posterior pole ‘p. pole” protocol which encompasses an area of 30 × 25° with 61 scans per section. Each scan was assessed for the presence of intraretinal cysts, subretinal fluid, fvPED, and drusen. Manual measurement of retinal thickness was performed in the center of the fovea.

To confirm the presence of IRN and CRA in each patient, the results of an OCTA scan performed on the same day as the SDOCT scan were analyzed. The OCTA examination was performed using a RTVue XR Avanti device (Optovue, Fremont, CA, USA) using The AngioVue™ Imaging System (RTVue XR Avanti, Optovue Inc., Fremont, CA, USA) ver. 2017.1.0.151 software. This system used a split-spectrum amplitude decorrelation angiography (SSADA) software algorithm and acquired 70,000 A-scans per second to compose OCTA volumes consisting of 400 × 400 A-scans. The A scan depth is 2.3 mm with an axial resolution in tissue of 5 μm and a transverse resolution in tissue of 15 μm. Orthogonal registration and merging of 2 consecutive scans were used to obtain OCTA macula volume scans of a central 3 × 3 mm or, for larger lesions, a 6 × 6 mm macula area. Only technically good studies achieving a scan quality index (SQ index) of 6 or more on a 10-point scale were analyzed. Each retinal section was independently assessed by two experts (J.J.K. and P.Z.) for the presence of IRN and CRA. IRN was distinguished on OCTA tomograms by the presence of pathological retinal flow located adjacent to hyperreflective foci within the outer retinal layers (Figure 1A). CRAs were presented as flow signal extending through the RPE (Figure 1B). Upon identifying IRNs and CRAs, their position on an en-face image was evaluated. Using commercial AngioVue software version 2017.1.0.151 the vessel densities of the superficial (SVC) and deep (DVC) vascular complexes was calculated as the percentage of area occupied by flowing blood vessels in the selected region. The lesion size and flow area within the neovascular membrane beneath the RPE were also measured.

### Statistical Analysis

Summary statistics for variables are presented as mean ± one standard deviation (SD). Differences between groups were assessed using the Student’s *t*-test for continuous variables. The χ2 test with Yates’ correction or Fisher’s exact test (depending on expected values in contingency tables) were used to calculate the differences between categorical variables. The odds ratio was used to quantify the association between presence of cysts and remaining variables with 95% confidence interval. To compare evaluation of OCTA scans by two different graders, intergrader agreement coefficient was calculated. The results were considered statistically significant when the *p*-value was less than 0.05.

## 3. Results

Forty-two eyes of forty patients were included in the study. In 2 cases, both eyes of the same patient were analyzed. Group 1 included 21 eyes from 12 males and 9 females. The mean age of patients in this group was 79.6 ± 7.12 years. There were 21 eyes from 11 males and 8 females in Group 2. Patients in Group 2 were significantly younger than those in Group 1 with a mean age of 72.5 ± 8.25 years (*p* < 0.01). Both groups included eyes previously treated with anti-VEGF agents. Group 1 had 7 eyes and Group 2 had 8 eyes receiving such treatment agents. The patient demographics of both study groups are presented in Table 1.

The best corrected visual acuity was significantly higher in Group 2 compared to Group 1 (0.57 ± 0.20 vs. 0.43 ± 0.21). On color fundus photography, hemorrhages were visible in 11 of the eyes (52.4%) in Group 1 and 2 eyes (9.5%) in Group 2. In the 11 eyes from Group 1 (52.4%), both subretinal fluid and intraretinal cysts were visible on SDOCT. Fluid under the sensory retina was present in all eyes in Group 2. The mean foveal thickness in Group 1 was 379.05 ± 120.10 µm compared to 325.11 ± 103.55 µm in Group 2. The OCTA study quality index was 7.42 ± 1.1 in Group 1 and 7.47 ± 1.12 in Group 2 (*p* = 0.895)

IRN was observed in 20 eyes (95.2%) and CRA in 18 (80.95%) eyes on at least one retinal cross section obtained by OCTA in Group 1. The IRN-like lesions were usually located adjacent to intraretinal cysts where hyperreflective foci were present on SDOCT. These lesions were usually accompanied by blurring of the boundaries between the different layers of the retina (Figure 2).

In Group 2, the presence of IRN was observed in 5 eyes (23.8%) and CRA in 16 eyes (76.2%). The abnormal flows described in Group 2 were located in areas devoid of subretinal fluid where the pigment epithelium covering the choroidal neovascularization was directly adjacent to the sensory retina (Figure 3). The IRNs in this group also occurred in the vicinity of hyperreflective foci. However, the retinal structural abnormalities were not as severe as in Group 1 and tracking of the individual retinal layers was usually possible (Figure 3). IRN or CRA-like lesions visible on retinal cross-sections usually corresponded to focal flows on en face OCTA images within the DVP or outer retina and were usually difficult to distinguish from physiological flows within the DVP (Figure 2). Statistical analysis shows that patients with cysts were more likely develop IRN (OR: 50.00; 95% CI: 5.43–460.52; *p* < 0.001). The association between presence of cysts and RCA is not statistically significant (OR: 1.33; 95% CI: 0.30–5.84; *p* > 0.05). Results are shown in Figure 4. The intergrader agreement coefficient between observers was 0.88 for the presence of IRN and 0.83 for the presence of CRA.

In Group 1, vascular flow on OCTA within the SVP and especially the DVP on en face imaging was characterized by the appearance of areas devoid of vessels where fluid-filled cysts were present. The mean vessel density of the SVP over the entire measurement area in Groups 1 and 2 were 43.11 ± 5.99% and 46.91 ± 5.20%, respectively (*p* < 0.05). For the DVP, this parameter yielded a value of 44.42 ± 4.65% in Group 1 and 45.30 ± 6.80% in Group 2 (*p* = 0.62). The foveal avascular zone (FAZ) area was not significantly different between the two groups (0.25 ± 0.10 mm^2^ in Group 1 and 0.30 ± 0.10 mm^2^ in Group 2).

Pathological flows in the outer retinal layers corresponding to MNVs located beneath the RPE (stage III RAP) were observed in 14 eyes (66.7%) from Group 1. The location of these flows wasclassified as subfoveal in 4 eyes, juxtafoveal in 6 eyes, and extrafovealin 4 eyes. The presence of serous PED or fluid under the sensory retina without pathological flows typical of MNV corresponding to stage II RAP occurred in 5 eyes (23.8%) from Group 1. MNV1 on OCTA examination was evident in all eyes in Group 2. In 15 eyes it had a subfoveal and in 4 eyes juxtafoveal location. Extrafoveal MNV was only observed in 2 eyes in this group. The most common MNV shape observed in Group 1 was glomerulus (7 eyes). In Group 2, the most common shapes were seafan (10 eyes) and medusa head (9 eyes). The mean area of pathological flows within the MNV was 0.44 ± 0.70 mm^2^ in Group 1 and 2.28 ± 2.66 mm^2^ in Group 2 (*p* < 0.01). All parameters measured during OCTA examination for both groups are presented in Table 2.

## 4. Discussion

This study compares the results of macular OCTA examinations performed in two groups of patients with exudative nAMD with differing results on SDOCT imaging. In the eyes belonging to Group 1, which had intraretinal cysts, RAP features on OCTA examination were seen in 90.5% of cases. In eyes classified into Group 2, fluid was present under the sensory retina on SDOCT and intraretinal cysts were not visible, changes indicative of IRN on OCTA were seen in 23.8% of eyes. There were fewer differences between groups in regard to the presence of blood flow abnormalities at the RPE-retinal sensory level specific to CRA. These changes occurred in 80.95% and 76.2% of eyes in Group 1 and Group 2, respectively.

Sensory retinal edema in the macula, in addition to hemorrhages, was one of the first signs of RAP observed on funduscopic examination [1,2,3]. On SDOCT retinal sections, the characteristic changes typical of RAP were intraretinal cysts, focal RPE damage, and intraretinal hyperreflective foci [5,6]. However, making a definitive diagnosis required the use of IA which allows visualization of retinal anastomoses and CRA [7,8,9]. The introduction of OCTA has greatly facilitated the diagnosis of RAP and reduced the performance of expensive and often difficult to access IA. OCTA can non-invasively detect abnormal flows within the sensory retina and CRA in 60 to 100% of eyes with RAP [10,11,12,13].

However, intraretinal cysts in the macula of patients with nAMD can have a variety of causes. We should not forget the possible coexistence of nAMD and diabetic macular edema, branch retinal vein occlusion, or macular telangiectasia. Vitreous macular traction or the presence of a preretinal membrane are also common causes of cyst formation. The use of SDOCT enables an accurate analysis of the individual retinal layers and demonstrates very high sensitivity in detecting even small amounts of fluid within the retina. The presence of such fluid is the most important component of MNV activity. Dark, hyporeflective serous fluid is usually easily distinguished from the more or less hyperreflective retinal layers allowing intraretinal cysts to be easily spotted during the examination. Our study confirmed that finding intraretinal cysts on SDOCT in patients with drusen or PED especially when accompanied by hyperreflective foci in the sensory retina is associated with a high probability of IRN on OCTA. However, the absence of cysts does not rule out the presence of IRN [14]. Pathological vascular flow within the sensory retina can occur in the early stages of IRN development without the presence of intraretinal cysts. In such cases, SDOCT shows only hyperreflective lesions sometimes passing through all retinal layers. This was the case in 23.8% of eyes in Group 2, where despite the absence of intraretinal cysts, the OCTA examination showed pathological flow within the macula.

The presence of CRA on fluorescein angiography is usually associated with the presence of advanced exudative nAMD with a significant degree of scarring [15]. OCTA showed the presence of pathological flows passing through the RPE in 74% of eyes with RAP [16]. This is relatively common and also occurred in the eyes of Group 2 in which the MNV was beneath the RPE and fluid accumulated only under the sensory retina. The fine flows passing through RPE observed in Group 2 may not be real anastomosis connecting the MNV vessels to the intraretinal circulation but rather correspond to neovascular membrane vessels that pass through the RPE to form the MNV located beneath the sensory retina (MNV2), which arises from the progression of MNV1.

Despite the appearance of IRN, retinal vessel density in Group 1 was not greater than in Group 2. In the SVP, we observed a reduction in vascular density due to the presence of cysts, which were recorded as areas devoid of vessels. The vessel density within the DVP in both groups was similar. This may have been influenced by the presence of IRNs offsetting the negative effect of intraretinal cysts on vessel density measurements.

Pathological flow under the RPE on OCTA was observed in all Group 2 eyes and in 71.4% of Group 1 eyes that developed stage 3 RAP. Despite the similar locations of the MNV beneath the RPE, these changes differed between groups. In Group 1, the area of the neovascular membrane was significantly smaller and the membrane itself was more likely to be in a perifoveal or extrafoveal location than in Group 2.

This study does have some limitations. Due to the small number of eyes meeting the inclusion criteria, patients treated with anti-VEGF agents as well as untreated patients were eligible for the study. However, this did not seem to have much effect on the results as previously treated and untreated eyes were found in similar proportions in both groups. Additional limitations exist specific to OCTA. An OCTA examination may fail to detect extremely slow blood flow and thus can miss early, small MNV3 lesions. In addition, OCTA results are strongly influenced by the quality of the examination and are susceptible to artefacts. Due to the similar SQ ratios in both groups and the use of the appropriate software version, the influence of these factors was probably minor.

## 5. Conclusions

The presence of intraretinal cysts on SDOCT retinal sections in eyes with nAMD features such as soft drusen, pseudodrusen, and MNV corresponds to the presence of IRN on OCTA examination. This is especially true if hyperreflective foci are present in the vicinity of the cysts which may facilitate the diagnosis of MNV3. However, the absence of a cyst does not exclude the presence of IRN and CRA which can be identified on OCTA.

## Figures and Tables

**Figure 1 medicina-58-00676-f001:**
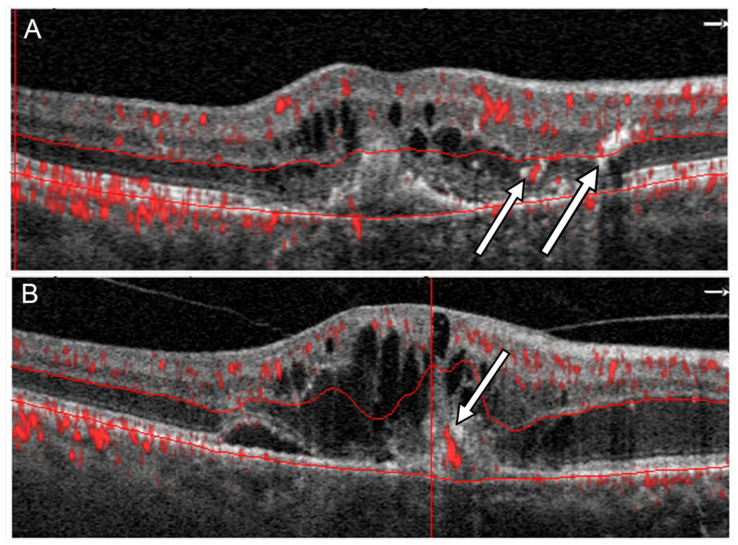
Two different patterns of flow in RAP in OCTA: (**A**) A flow signal in outer retina in vicinity of hyperreflective lesions corresponding to IRN (arrows), (**B**) a flow signal passing through the RPE corresponding to CRA (arrow).

**Figure 2 medicina-58-00676-f002:**
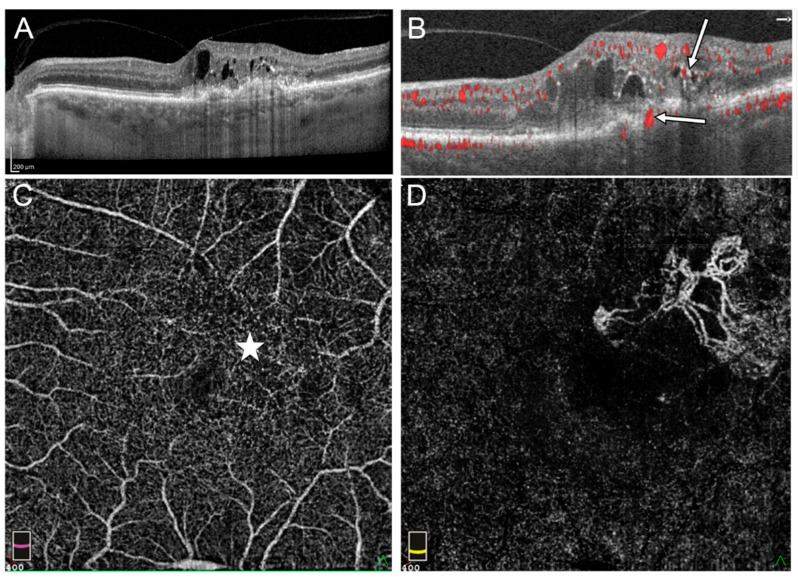
Stage III RAP in the left eye: (**A**) OnSDOCT cross-section, the intraretinal cysts and hyperreflective foci in outer retinal layers can be observed, (**B**) OCTA cross section reveals flows within sensory retina corresponding to IRN (thin arrow) and the flow passing RPE (thick arrow), (**C**) en face OCTA on the level of the deep vascular plexus (DVP) demonstrated irregular area of flow in superior temporal part of macula where intraretinal cysts are located (star), (**D**) in en face OCTA, flows of extrafoveal MNV on the level of external retina are visible.

**Figure 3 medicina-58-00676-f003:**
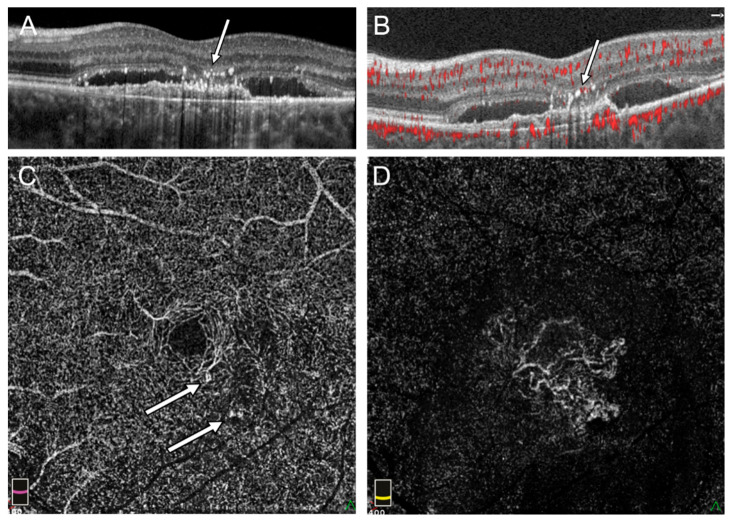
Flow signal in outer retina in the left eye with subretinal fluid and no sign of intraretinal cysts: (**A**) Hyperreflective foci above fibrovascular PED and fluid under sensory retina in SDOCT cross section (arrow), (**B**) pathologic flow adjacent to hyperreflective foci in OCTA (arrow), (**C**) en face OCTA on the level of DVP shows focal points of increased flow (arrow), (**D**) the external retina with flow of subfoveal MNV in en face OCTA.

**Figure 4 medicina-58-00676-f004:**
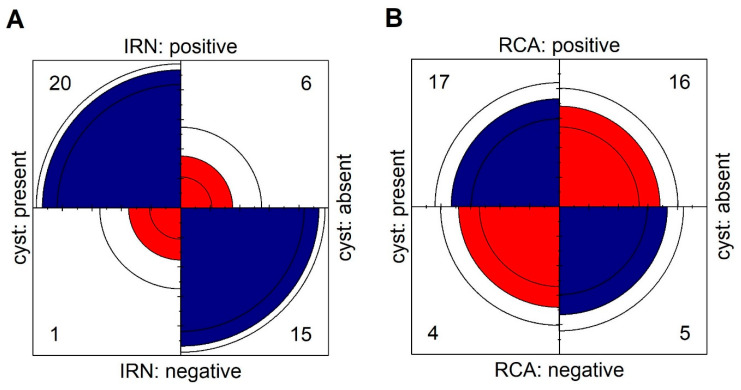
Fourfold display for presence of cysts and (**A**) IRN, (**B**) RCA. The area of each quadrant is proportional to the cell frequency. Circular arcs show the limits of a 95% confidence interval for the odds ratio in logarithmic scale.

**Table 1 medicina-58-00676-t001:** Demographic data and basic parameters describing examined groups.

	Group 1	Group 2	*p*-Value *
Number of eyes (n)	21	21	
Number of patients (n)	21	19	
Woman:man ratio	12:9	11:8	
Average age (years)	79.57 ± 7.12	72.63 ± 8.25	*p* < 0.01
Primarily treated eyes with anti-VEGF agents (n)	7	8	
Treatment naive eyes (n)			
Average BCVA (Snellen)	14	13	
Average macular thickness (µm)	0.43 ± 0.21	0.57 ± 0.20	*p* < 0.05
	379.05 ± 120.10	325.10 ± 103.55	*p* < 0.01

* Independent *t*-test, *p*-value < 0.05 was considered to be statistically significant. Abbreviations: VEGF, vascular endothelial growth factor; BCVA, best-corrected visual acuity.

**Table 2 medicina-58-00676-t002:** Comparison of quantitative parameters of SVP, DVP and MNV in OCTA between groups.

	Group 1	Group 2	*p*-Value *
Density SVP (%)			
- whole image	43.11 ± 5.99	46.91 ± 5.20	<0.05
- fovea	25.75 ± 6.70	22.99 ± 8.09	NS
- parafovea	44.96 ± 5.99	47.55 ± 7.45	NS
Density DVP (%)			
- whole image	44.42 ± 4.65	45.30 ± 6.81	NS
- fovea	34.09 ± 7.99	29.82 ± 8.78	NS
- parafovea	46.05 ± 4.96	47.13 ± 7.60	NS
FAZ area (mm^2^)	0.250 ± 0.10	0.300 ± 0.10	NS
Lesion area MNV (mm^2^)	1.50 ± 1.97	4.61 ± 5.47	<0.05
Lesion flow area MNV (mm^2^)	0.56 ± 0.70	2.28 ± 2.66	<0.01

* Independent *t*-test, *p*-value < 0.05 was considered to be statistically significant. Abbreviations: SVP, superficial vascular plexus; DVP, deep vascular plexus; FAZ, foveal avascular zone; MNV, macular neovascularization; NS, not significant.

## Data Availability

The data presented in this study are available on request from the corresponding author. The data are not publicly available due to privacy concerns.

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
