# Peer review of "Intraretinal Cysts as a Manifestation of Retinal Angiomatous Proliferation in Optical Coherence Tomography Angiography"

_medicina, 2022, doi:10.3390/medicina58050676_

Round 1
Reviewer 1 Report
This paper describes the use of OCTA to correlate nAMD features with sdOCT without FA. These results are interesting but a few points need to be clarified.
In the abstract the results and meaning behind them could be clarified i.e.
P1 L12: “The aim of the study was to determine if the presence of intraretinal cysts is synonymous with diagnosis of retinal angiomatous proliferation” and again in P2 L56 synonymous could be better stated as “positively correlated” or “consistent” or “associated”
P1 L21: “However, the absence of a cyst does not exclude the presence of IRN and CRA which can be identified on OCTA.” The information presented is valuable but needs a clear framing of the purpose of the study is here so “The results indicate that… therefore, … “
P2 L46 “However, in 46 many cases, the analysis of retinal lesions seen on spectral domain optical coherence tomography (SdOCT) is sufficient to make the initial diagnosis” This statement undermines the need for OCTA, unless stated OCTA is predictive or enhances the diagnosis
P2 L95: “The vessel densities of the superficial (SVC) and deep (DVC) vascular complexes 95 were measured using commercial software.” What software was used here, and what metric of vascular density was used?
In the statistical section please explain what techniques were used and what was the test conditions. Specifically, what is the control group for comparison.
P8 L236: “Despite the appearance of IRN, retinal vessel density in Group 1 was not greater than in Group 2. In the SVP, we observed a reduction in vascular density due to the presence of cysts, which were recorded as areas devoid of vessels.” Is vascular density a good discriminant feature, if not what may be a meter metric from this study?
Reviewer 2 Report
The authors did a retrospective study on the data collected with spectral-domain optical coherence tomography (SD-OCT) and Optical coherence angiography (OCA) on the age-related macular degeneration patients treated with Anti-vascular endothelial growth factor therapy. They conclude the presence of intraretinal cysts in SD-OCT data of patients most likely will have the intraretinal neovascularization (IRN). This study will be useful to non-invasively detect retinal revascularization, unlike invasive indocyanine green angiography. This work is suitable for publication nevertheless this manuscript can be accepted after addressing the below minor corrections.
- In the abstract several words are abbreviated it is recommended to write their full forms for instance
- in line 11 Sd OCT
- in line 12, AMD
- in line 14, OCTA
- Recommended to add references to the following sentence in lines 41-43 “In stage III of the disease, the pathological vessels under the RPE are observed together with the formation of a chorioretinal anastomosis (CRA).”
- In general, the optical coherence tomography imaging community abbreviates spectral-domain optical coherence tomography as SDOCT, authors may replace the word SdOCT with SDOCT. (line 48).
- In line 61, I did not notice the anywhere full form of anti-VEGF, I suggest writing the full form once.
- Line 62, I did not understand what is CM UMK stands for?
- In line 68, Is the author means to say -5.0 Diopters?
- In line 86, I recommend authors write the important imaging specifications of RTVue XR Avanti Device (Optovue, USA), like lateral resolution, axial resolution imaging speed etc.
- In lines 95-96, the authors stated, “The vessel densities of the superficial (SVC) and deep (DVC) vascular complexes were measured using commercial software.” If authors write the name of the software will be useful for the readers.
- In line 124-125, “The OCTA study quality index was 7.42 ± 1.1 in Group 1 and 7.47 ± 1.12 in Group 2. ”, what is the significance of the quality index here?
- In line 156, The word DVP is not expanded so far, I feel too many abbreviations are used in the manuscript some readers will be lost, I recommend checking the abbreviations in the entire manuscript.
Round 2
Reviewer 1 Report
In review of the changes I have no further comments.
Author Response
Dear Reviewer,
Thank you for comments that contributed to the improvement of our article.
Reviewer 2 Report
The authors addressed all the recommended changes but the authors did not completely address my previous Comment 7. I recommend writing axial and lateral resolutions of those systems those are important parameters to resolve. This Manuscript may be accepted once they add the above content.
one more general suggestion, if authors paste the content modified in the manuscript in the author's reply document will help the reviewer for quick review.
Author Response
We fully agree with this comment and we clarified this part of the manuscript according to suggestions: "The A scan depth is 2.3 mm with an axial resolution in tissue of 5 μm and a transverse resolution in tissue of 15 μm".
"[...] The OCTA examination was performed using a RTVue XR Avanti device (Optovue, USA) using AngioVue ver. 2017.1.0.151 software. This system used a split-spectrum amplitude decorrelation angiography (SSADA) software algorithm and acquired 70.000 A-scans per second to compose OCTA volumes consisting of 400 × 400 A-scans. The A scan depth is 2.3 mm with an axial resolution in tissue of 5 μm and a transverse resolution in tissue of 15 μm. Orthogonal registration and merging of 2 consecutive scans were used to obtain OCTA macula volume scans of a central 3 × 3 mm or, for larger lesions, a 6 × 6 mm macula area. [...]"